**PLOS** NEGLECTED TROPICAL DISEASES

# Latent Class Analysis: Insights about design and analysis of schistosomiasis diagnostic studies

Artemis Koukounari[1,2]*, Haziq Jamil[3], Elena Erosheva[4], Clive Shiff[5], Irini Moustaki[6]

1 Product Development Personalized Health Care, F. Hoffmann-La Roche Ltd., Welwyn Garden, United Kingdom, 2 Department of Infectious Disease Epidemiology, London School of Hygiene & Tropical Medicine, London, United Kingdom, 3 Mathematical Sciences, Faculty of Science, Universiti Brunei Darussalam, Bandar Seri Begawan, Brunei, 4 Department of Statistics, School of Social Work, Center for Statistics and the Social Sciences, University of Washington, Seattle, Washington, United States of America, 5 Molecular Microbiology and Immunology Department, John Hopkins Bloomberg School of Public Health, 6 Department of Statistics, London School of Economics and Political Science, London, United Kingdom

* Artemis.Koukounari@roche.com

## Abstract

Various global health initiatives are currently advocating the elimination of schistosomiasis within the next decade. Schistosomiasis is a highly debilitating tropical infectious disease with severe burden of morbidity and thus operational research accurately evaluating diagnostics that quantify the epidemic status for guiding effective strategies is essential. Latent class models (LCMs) have been generally considered in epidemiology and in particular in recent schistosomiasis diagnostic studies as a flexible tool for evaluating diagnostics because assessing the true infection status (via a gold standard) is not possible. However, within the biostatistics literature, classical LCM have already been criticised for real-life problems under violation of the conditional independence (CI) assumption and when applied to a small number of diagnostics (i.e. most often 3-5 diagnostic tests). Solutions of relaxing the CI assumption and accounting for zero-inflation, as well as collecting partial gold standard information, have been proposed, offering the potential for more robust model estimates. In the current article, we examined such approaches in the context of schistosomiasis via analysis of two real datasets and extensive simulation studies. Our main conclusions highlighted poor model fit in low prevalence settings and the necessity of collecting partial gold standard information in such settings in order to improve the accuracy and reduce bias of sensitivity and specificity estimates.

## Author summary

Accurate schistosomiasis diagnosis is essential to assess the impact of large scale and repeated mass drug administration to control or even eliminate this disease. However, in schistosomiasis diagnostic studies, several inherent study design issues pose a real challenge for the currently available statistical tools used for diagnostic modelling and associated data analysis and conclusions. More specifically, those study design issues are: 1) the

**Data Availability Statement:** Within the shiny application, data can be viewed when response patterns are displayed.

**Funding:** The authors received no specific funding for this work.

**Competing interests:** The authors have declared that no competing interests exist.

inclusion of small number of diagnostic tests (i.e. most often five), 2) non formal consensus about a schistosomiasis gold standard, 3) the contemporary use of relatively small sample sizes in relevant studies due to lack of research funding, 4) the differing levels of prevalence of the studied disease even within the same area of one endemic country and 5) other real world factors such as: the lack of appropriate equipment, the variability of certain methods due to biological phenomena and training of technicians across the endemic countries because of scarce financial resources contributing to the existing lack of a schistosomiasis gold standard. The current study aims to caution practitioners from blindly applying statistical models with small number of diagnostic tests and sample sizes, proposing design guidelines of future schistosomiasis diagnostic accuracy studies with recommendations for further research. While our study is centred around the diagnosis of schistosomiasis, we feel that the recommendations can be adapted to other major tropical infectious diseases as well.

## Introduction

Schistosomiasis affects over 250 million people in rural and some urban populations across much of Africa and regions in South America, the Caribbean, People's Republic of China, and Southeast Asia [1, 2]. Since the new millennium, schistosomiasis interventions are escalating [3] with elimination and interruption of transmission in selected areas to be set as World Health Organization key goals for 2025. Accurate schistosomiasis diagnosis is essential to assess the impact of large scale and repeated mass drug administration to control or even eliminate this disease [4]. Some molecular assays could constitute the gold standard for schistosomiasis diagnosis [5, 6] but the lack of sufficient funds, suitably trained staff and laboratory supplies still hinders their use in endemic countries. Certainly, with the African schistosomes, some work has been done to answer in the affirmative, the detection of species specific DNA in body fluids, either blood or urine. Lodh et al. [7] carried out a study in Zambia to compare KK, CCA and DNA (from urine) in an exposed population in Zambia. A total of 89 volunteers were examined, KK showed 45/89 infected, CCA 55/89 (6 false positives) and 79/89 had detectable *Schistosoma mansoni* DNA. Clearly if there is a schistosomiasis gold standard, it is the presence of parasite specific DNA. For *Schistosoma haematobium* work has been done in Western Nigeria [8], comparing the presence of parasite eggs detected on the filter paper from the filtered urine, haematuria detected by dipstick, and detection of the Dra1 DNA extracted from the filters. Some 400 specimens were examined in that study and the conclusion was that detection of Dra1 fragment is a definitive test for the presence of *S. haematobium* infection.

Nevertheless, renewed emphasis upon research and evaluation of schistosomiasis diagnostic tools has been generated [5, 9] with a notable increase of diagnostic studies particularly during the last five years [10–27], using a 2-latent class model without a gold standard [28, 29] to estimate and evaluate diagnostic accuracy.

Latent class models (LCMs) have been increasingly utilised to describe the relationship among schistosomiasis diagnostic tests and other infectious diseases as being indicators of latent classes representing usually two states of a disease that of infected and non-infected [30]. Such models explain the associations among a number of diagnostic tests or observed items in general using a small number of unobserved classes. van Smeden et al. [30] reviewed a large number of publications in terms of their LCA methodology as a diagnostic tool in the absence of gold standard.

The simplest LCM assumes diagnostic tests are independent given the latent class. This is known as the conditional independence (CI) assumption implying that if the true disease status is misclassified by one test, the probability that it will be misclassified by another test will not be affected. However, as with any statistical model, the validity of modelling results in the real world is often jeopardised, leading to systematically overestimated classification accuracy rates [31] when assumptions are violated. Hence, checking model assumptions through overall goodness-of-fit tests such as chi-square or likelihood ratio tests and measures of fit such as residuals constitute an essential part of the data analysis. This is important in order to appraise the validity of the reported results when LCMs without a gold standard are used.

Here we start by discussing violations of the CI assumption in general and within the context of schistosomiasis diagnosis in particular. LCMs have already been criticised for real-life problems not only on the ground of violation of the CI assumption but also when applied to a small number of diagnostic tests [32]–another pertinent problem in schistosomiasis diagnostic studies. We subsequently outline two more variants of LCMs with conditional dependencies and introduce the idea of including the partial gold standard for the improvement of bias and precision of these model parameter estimates. Currently there is no agreed gold standard diagnostic test for schistosomiasis; however, we are assuming that in due course the presence of parasite specific DNA in the urine will be acceptable as the gold standard. This is based on evidence from two previous studies [7, 8] as well as a more recent research by Archer et al. [33]. Another practical solution for including another partial 'gold standard' could be to combine information on a subset of individuals from different tests as in [34] and compare those to data from other imperfect diagnostic tests on all individuals via the proposed models. Furthermore, we fit these three variations on LCMs and discuss results on real data for *S. haematobium* infection from Ghana [24] and *S. mansoni* infection from Uganda [10]. Both of these datasets contain five observed binary items, have similar sample sizes and lack gold standard evaluations. Such datasets are typical examples of the majority of diagnostic studies to date and constitute a good basis for the current study to build simulations and provide recommendations for correct interpretation of their results. Simulations on relevant scenarios for *S. mansoni* infection allows us to study the performance of the classical LCM when CI is violated. Within these simulations, we also examine whether partial gold standard information improves the accuracy of parameter estimates (prevalence, sensitivities and specificities for the examined diagnostic tests) in the presence of model violations. Such a process permits to examine how would our results (bias and precision) change if gold standard information was available for some individuals for the three fitted latent variable models.

We aim to caution the practitioner not to blindly apply methods for estimating diagnostic errors particularly with small sample sizes and limited number of diagnostic tests (i.e. five) in schistosomiasis diagnostic studies. We would also like to emphasise at this point that the current study does not aim necessarily to promote or demote particular available diagnostic tests. With transparency (i.e. clearly describing the model assumptions, structure and parameter values) as well as validation (i.e. subjecting the models to tests such as comparing the model's results with events observed in reality through real world data analysis and simulations) the current study aims to show under which scenarios LCMs might work or not for the evaluation of existing schistosomiasis diagnostics. Finally, we propose guidelines for design of future schistosomiasis and other infectious diseases diagnostic accuracy studies and make recommendations for further research. We additionally provide the JAGS code used to fit the discussed models in this study in the S1 File. We can also share codes for simulations upon request for other interested researchers.

### Assumption of conditional independence in the 2-latent class model

The CI assumption might be plausible when different tests are based on different scientific/technological grounds or when they measure different characteristics of the disease. However, this assumption often fails in practice. When some individuals without the disease of interest have another condition in common that increases the likelihood of two (or more) tests to render false positives because they are based on a comparable biological principle [35]–this can induce correlation between tests beyond the one explained by the disease status of interest. This could be the case in patients with alternative causes of microhaematuria and not infected with *S. haematobium* and perhaps with diagnostic tests such as reagent strips and urine filtration of 10 ml or whole urine samples from single urine specimens [36].

The existence of some other disease that has two or more diagnostic tests in common with the disease of primary interest can violate the CI of the examined variables even within latent classes of the disease of primary interest [29]. More precisely, again within the context of diagnosis of urogenital *S. haematobium* infection, a series of self-reported urogenital symptoms (i.e. abnormal discharge colour, abnormal discharge smell, burning sensation in the genitals, bloody discharge, genital ulcer, red urine, pain on urination, stress incontinence and urge incontinence) might be also caused by sexually transmitted diseases such as *Chlamydia trachomatis* [21]. Individuals can be co-infected with *C. trachomatis* and urogenital *S. haematobium* at the same time and such symptoms might be also used for diagnosis of *C. trachomatis* in schistosomiasis endemic countries.

Another cause for conditional dependence among test results could arise if there is a subgroup of individuals with an early or less severe stage of the disease of interest and if these individuals are more likely to be missed by some tests [37]. The latter can be the case in *S. mansoni* infection where the parasitological Kato-Katz (KK) method [38] to detect Schistosoma eggs in stool exhibits day-to-day variation in faecal egg output, and has low sensitivity in detecting light-intensity infections [12]. In addition, serological tests, such as screening for antischistosomal antibodies, are of limited use for the diagnosis of active infection, as large parts of the population may carry antibodies due to past cured infections [5, 9]. The need to understand the mechanisms of the tests, and in particular their mutual dependence in diseased and non-diseased subjects, as well as a clear clinical definition for disease has been already previously highlighted [39].

Finally, an additional consideration is if some of the tests are administered by the same nurse or doctor and some are self-reported, there might be a method-effect and the outcomes might not be solely explained by the disease status.

## Material and methods

### Statistical models with conditional dependencies

Two more variants of LCMs that are commonly used in epidemiological studies to define disease status in the absence of a gold standard and when the CI assumption does not hold, are the latent class random effect model (LCRE), and the finite mixture latent class model (FM). In the LCRE, conditional dependencies among variables is accommodated through an individual-specific random effect applied to all diagnostic tests capturing heterogeneities that cannot be explained by the two classes. In the current study we do not examine latent class models with random effects to specific diagnostic items as this would imply that we have pre-knowledge on which diagnostic items are correlated beyond the correlation that can be explained by the latent structure. When subjects have the disease or are healthy with certainty, in the data this will be seen as an excess number of all 0 or all 1 test results. Data that contain an excess

number of zeros, relatively to what the model allows/predicts, are called zero-inflated data [40]. In the case of schistosomiasis, it is well known that schistosomes are over dispersed within certain populations and age-groups; a small number of individuals carry most of the parasites and thus the all-zero effect is most relevant [41]. The FM allows for the modelling of excess zeros (all-zero) and excess ones (all-one) in the data as separate components of a LCM. Mathematical details of the LCM with CI, LCRE and FM are given below.

**Latent class model with conditional independence.** We first discuss the standard LCM with CI. The marginal probability of a response pattern $r$ is given by:

$$P(\mathbf{y}_r) = \sum_{j=0}^{1} P(\xi = j)P(\mathbf{y}_r \mid \xi = j), \quad r = 1, \ldots, R, \tag{1}$$

where $\tau = P(\xi = 1)$ defines the disease prevalence parameter for the population of interest. Furthermore, the tests are assumed to be conditionally independent given the true disease status which implies: $P(\mathbf{y}_r \mid \xi = j) = \prod_{i=1}^{p} P(y_i \mid \xi = j)$. Each $y_i$ in turn is modelled with the Bernoulli distribution with class conditional probabilities $\lambda_{1i} = P(y_i = 1 \mid \xi = 1)$ and $\lambda_{2i} = P(y_i = 1 \mid \xi = 0)$. The model parameters are the $\lambda_{1i}, \lambda_{2i}, \tau$, for $i = 1, \ldots, p$. Furthermore, for the $i$th diagnostic test, its sensitivity is the probability of the positive test result given that the true diagnosis is positive, $P(y_i = 1 \mid \xi = 1)$, and its specificity is the probability of a negative response given that the true diagnosis is negative, $P(y_i = 0 \mid \xi = 0) = 1 - P(y_i = 1 \mid \xi = 0)$. The sensitivity and specificity of test $i$ implied by the latent class model are then simply $P(y_i = 1 \mid \xi = 1) = \lambda_{1i}$ and $P(y_i = 0 \mid \xi = 0) = 1 - \lambda_{2i}$.

**Latent class Gaussian random effects model.** In some applications the CI assumption does not hold for all or some of the indicators. To accommodate conditional dependencies among variables, an individual-specific random effect is introduced to capture heterogeneities that cannot be explained by the two classes [42, 43]. The random effect is a continuous, normally distributed, unobserved variable that serves as a summary of individual characteristics that explain–together with the disease status–the outcome of a test. This is known as a latent class Gaussian random effects model (LCRE). Under this model, the $y_i$ are modelled with the Bernoulli distribution with $P(y_i = 1 \mid \xi = j, u) = \Phi(\beta_{ij} + \sigma_j u)$, where $\Phi$ is the cumulative distribution function of the normal and $u$ is the individual-specific random effect that follows a standard normal distribution. The parameter $\sigma_j$ allows the random effect to have a different variance in each latent class defined by $\xi$. The marginal probability of a response pattern $r$ under CI is given by:

$$P(\mathbf{y}_r) = \sum_{j=0}^{1} P(\xi = j) \int \prod_{i=1}^{p} P(y_i \mid \xi = j, u)\phi(u)du, \quad r = 1, \ldots, R. \tag{2}$$

The sensitivities and specificities for the latent class Gaussian random effects model for each test $i$ are given in closed form:

$$P(y_i = 1 \mid \xi = 1) = \Phi\left(\frac{\beta_{i1}}{(1 + \sigma_1^2)^{1/2}}\right)$$

and

$$P(y_i = 0 \mid \xi = 0) = 1 - \Phi\left(\frac{\beta_{i0}}{(1 + \sigma_0^2)^{1/2}}\right)$$

respectively. When $\sigma_0^2 = \sigma_1^2 = 0$, the model reduces to the latent class model with local

independence. Although such a model does not specify (in)dependence among certain tests, we would expect them to capture such structures when these models provide good fit to the data overall. The sensitivities and the specificities computed after Eq (2) depend on the variance of the random effect which is allowed to be different in each class but common for all diagnostic tests.

**Finite mixture latent class model.** The above two models do not account for subjects who either have the disease or are healthy with certainty. As already discussed, in the data this will be seen as an excess number of all 0 or all 1 test results. Ignoring that excess in the data can affect the true dimensionality of the data detected by standard goodness of fit tests and can also distort the parameter estimates and as a result sensitivities and specificities.

The finite mixture latent class model allows for the modelling of excess zeros (all-zero) and excess ones (all-one) in the data as separate components of a latent class model. To model all-zero and all-one effects, the finite mixture model [32] (FM) is employed. The model uses the two-class structure as its basis and adds two point masses for the combinations of all-zero and all-one responses. These point masses correspond to the healthiest and the most severely diseased patients that are always classified correctly. This model can be also considered as a latent class model with four classes, of which two classes are fitted as having a point mass. In the truly diseased class the probability of a positive outcome test is 1 and in the truly healthy class the probability of a positive outcome is 0. Let $t$ be an indicator that denotes correct classification. Specifically, let $t = 0$ if a healthy subject is always classified correctly (i.e., has the all-zero response pattern with $p$ tests), $t = 1$ if a diseased subject is always classified correctly, and let $t = 2$ otherwise. Thus, subjects are either always classified correctly, when either $t = 0$ or $t = 1$, or a diagnostic error is possible when $t = 2$. Denote the probabilities for correctly classifying diseased and healthy subjects by $\eta_1 = P(t = 1)$ and $\eta_0 = P(t = 0)$, respectively. Let also $w_i(\xi)$ denote the probability of the $i$th test making a correct diagnosis when $t = 2$. The finite mixture model of Albert & Dodd [32] assumes that the test results $y_i$ are independent Bernoulli random variables, conditional on the true disease status and the classification indicator. The model becomes:

$$P(y_i = 1 \mid \xi, t) = \begin{cases} w_i(1), & \text{if } \xi = 1 \text{ and } t = 2 \\ 1, & \text{if } \xi = 1 \text{ and } t = 1 \\ 1 - w_i(0), & \text{if } \xi = 0 \text{ and } t = 2 \\ 0, & \text{if } \xi = 0 \text{ and } t = 0. \end{cases} \quad (3)$$

Note that $P(y_i = 1 \mid \xi = 1, t = 0) = P(y_i = 1 \mid \xi = 0, t = 1) = 0$. The sensitivity and specificity of the $i$th test under the finite mixture model are then $P(y_i = 1 \mid \xi = 1) = \eta_1 + (1 - \eta_1)w_i(1)$ and $P(y_i = 0 \mid \xi = 0) = \eta_0 + (1-\eta_0)w_i(0)$, respectively.

The marginal probability of a response pattern $r$ is given by:

$$P(\mathbf{y}_r) = \sum_{t=0}^{2}\sum_{j=0}^{1} P(\mathbf{y}_r \mid \xi = j, t)P(\xi = j \mid t)P(t), \quad r = 1, \ldots, R, \quad (4)$$

Note that $P(\xi = 0 \mid t = 0) = 1$ and $P(\xi = 1 \mid t = 1) = 1$.

## An alternative solution: Inclusion of partial gold standard

Within the biostatistics literature, authors studied the effect of CI violation on LCM, LCRE and FM on parameter estimates [42–45] under different simulation scenarios. A review of some of the approaches can be also found in [46].

The main findings were that estimates of sensitivity, specificity, and prevalence were substantially different under the different types of models. Another important finding has been that when the number of diagnostic tests is less or equal to ten, it is typically very difficult to discern statistically between the two forms of conditional dependence. They comment that it may be easier to distinguish between these models in a larger number of tests and sample sizes. However, they do also recognise that in most cases the inclusion of 10 diagnostic tests is unrealistic which is definitively also the case of schistosomiasis.

Dendukuri et al. [47] discuss sample size calculations when two conditionally independent diagnostic tests are used in the absence of a gold standard. van Smeden [48] in his PhD thesis (Chapter 5) provides a closer examination of the models and simulation scenarios discussed by Albert & Dodd [32]. Erosheva & Joutard [49] performed simulation studies to examine recovery of diagnostic accuracy estimates confirming the findings of Albert & Dodd [42]. They identified the same difficulties in distinguishing the latent dependencies structure and biased estimations of diagnostic accuracy parameters when deterministic mixture components are present for the all-zero and all-one response patterns.

While these results would caution against using these LCMs, the difficulties of obtaining gold standard verification especially for infectious diseases in endemic countries remain a practical reality. Albert & Dodd [50] proposed a solution that collects gold standard information on a subset of subjects but incorporates information from both the verified and non-verified subjects during LCMs estimation, offering the potential for more robust LCM estimates; they conducted simulations assuming common sensitivity and specificity across six diagnostic tests.

In this article, inspired by Albert & Dodd [50], we conduct simulations by considering three latent structure models (LCM, LCRE and FM) and different proportions of gold standard for the evaluation of imperfect schistosomiasis diagnostics. However, unlike Albert & Dodd [50], we consider different sensitivities and specificities for different diagnostic tests and a range of prevalence settings, more appropriate in the context of schistosomiasis.

## Goodness-of-fit test statistics and measures of fit

In this section we address the following questions: 1) does the model (LC, LCRE or FM) fit the data? 2) which of the three models (LC, LCRE, FM) provide a better fit to the data and when? and 3) which diagnostic tests are not fitted well by the hypothesised model? A model that does not fit the data would imply that more classes might be needed to explain the dependencies and therefore lack of CI of the hypothesised model. In addition, chi-square type measures (bivariate residuals) for pair of diagnostic tests can also identify those items that are not fitted well by the hypothesised model and violate the CI assumption [51].

The first way for checking the fit of the models is to use global goodness-of-fit tests that compare the observed and expected (under the model) frequencies across the response patterns such as the likelihood ratio or Pearson chi-squared goodness-of-fit tests. The Pearson chi-squared goodness-of-fit test statistic, $X^2$, is given by:

$$X^2 = \sum_{r=1}^{2^p} N \frac{(p_r - \hat{p}_r)^2}{\hat{p}_r},$$

(5)

where $r$ represents a response pattern, $N$ denotes the sample size, and $p_r$ and $\hat{p}_r$ represent the observed and expected probabilities, respectively, of response pattern $r$. By multiplying $p_r$ and $\hat{p}_r$ by $N$ we obtain the observed and expected frequencies of pattern $r$. $\hat{p}_r$ for LC, LCRE and FM is estimated from $P(\mathbf{y}_r)$ given in Eqs (1), (2) and (4) respectively.

If the model holds, (5) is distributed approximately as $\chi^2$ with degrees of freedom equal to the number of different response patterns ($2^p$) minus the number of independent parameters (prevalence, sensitivities, specificities, variances of random effects) minus one. For the LC and LCRE with two latent classes, the degrees of freedom are $2^P - 2p - 2$ and $2^P - 2p - 4$ respectively. For the FM model (with four classes) the degrees of freedom are $2^P - 2p - 3$. The data can be considered as a $2^p$ contingency table. For instance, in our two real datasets from Ghana and Uganda, there are five diagnostic tests in each of them that give a $2 \times 2 \times 2 \times 2 \times 2$ contingency table. The sample sizes for the two datasets are 220 and 258 respectively and only 25 and 17 out of the total 32 response patterns appear in the sample and many patterns that occur in the sample have low (less than 5) frequencies. Theory tells us that small expected cell frequencies (less than five) (known also as sparseness) have a distorting effect on the chi-square tests. Under sparseness, the test statistic will no longer have the chi-square distribution and so from the practical point of view these tests cannot be used (see e.g. [52]).

Apart from looking at the whole set of response patterns, a second additional way could involve instead computing likelihood ratio and chi-square values for the two-way cross-tabulation of the diagnostic tests. That is, we can construct the $2 \times 2$ contingency tables obtained by taking two diagnostic tests at a time. For each cell of these bivariate contingency tables, we define a GF-Fit for which we compare the observed frequency with the expected frequency estimated under the corresponding latent variable model (LC, LCRE and FM). We use these terms to emphasise that there are no $\chi^2$-distribution associated with them. For category $a$ of variable $i$ and category $b$ of variable $j$ the GF-Fits are defined as follows

$$\mathrm{GF} - \mathrm{Fit}_{ab}^{(ij)} = N(p_{ab}^{(ij)} - \hat{p}_{ab}^{(ij)})^2 / \hat{p}_{ab}^{(ij)} \ . \tag{6}$$

As a rule of thumb, if we consider the $\mathrm{GF} - \mathrm{Fit}_{ab}^{(ij)}$ as having a $\chi^2$ distribution with one degree of freedom, then a value of those fit measures greater than 4 is indicative of poor fit at the 5% significance level [51, 53]. Similarly, summing these measures over $a$ and $b$ give the bivariate GF-Fits for variable $i$ and $j$ and therefore a value greater than sixteen will be then indicative of poor fit. The probabilities $\hat{p}$ can be evaluated under the LC, LCRE and FM models. The chi-squared residuals provide a measure of the discrepancy between the observed and the predicted frequency. A study of the bivariate chi-squared residuals provides information about where the model does not fit or in other words pair of variables for which the CI assumption is violated (local dependencies).

Furthermore, we conducted a parametric bootstrapping for finding the empirical distribution of the $\chi^2$ test statistic and the corresponding $p$-values. For each model, we generated $K = 10,000$ data sets from using the estimated model parameters from the data as the true ones. For each generated data set we computed the chi-square test statistic. To obtain the empirical $p$-value we count the number of times the test statistic from the original data is larger than the generated $\chi^2$ test statistics. We will illustrate this in both of our real datasets.

## Results

### Results from real datasets

We first used data from a study conducted in three villages northwest of Accra in Ghana which examined 220 adults using five *S.haematobium* diagnostic measures: microscopic examination of urine for detection of *S. haematobium* eggs, dipsticks for detection of haematuria, tests for circulating antigens, serological antibody tests (ELISA) and ultrasound scans of the urinary system [24]. We also used data from a most recent study from 258 children near Lake Albert in Uganda using four *S. mansoni* diagnostic measures: microscopy of duplicate Kato-

Katz smears from two consecutive stools (these are counting as two observed tests), urine-circulating cathodic antigen (CCA) dipstick, DNA-TaqMan and soluble egg antigen enzyme-linked immunosorbent assay (SEA-ELISA) [10]. De-identification of data before release to the statisticians and previous consent of study subjects was ensured. As mentioned in the introductory section, both datasets have five observed binary variables and do not have any observation with known gold standard. We employed Bayesian estimation methods to fit the models, assuming uninformative priors on the model parameters.

For the dataset from Ghana, all three models perform similarly in terms of model fit (the overall $X^2$ statistic is 12.79, d.f. = 20 for the LC model, 12.42, d.f. = 19 for the FM model and 13.23, d.f. = 18 for the LCRE model) and provide us with similar estimates of prevalence, sensitivity and specificity. The estimated parameters under the LC, LCRE and FM models as well as goodness-of-fit statistics are provided in Tables 1 and 2 and also in the Shiny application (https://perma.cc/QJ4V-GQ84) under the Ghana example tab. All the bivariate chi-squared residuals have values smaller than 4 indicating also a good fit. In addition, the empirical $p$-values for this data set also indicate a good fit (LC: $X^2$ = 16.42, $p$ = 0.982; LCRE: $X^2$ = 17.08, $p$ = 0.965 and FM: $X^2$ = 16.60, $p$ = 0.982).

We note that the prevalence level as estimated is fairly small, around 15%. We also note that sensitivity estimates for all diagnostic tests, except for ELISA, have very high associated uncertainty (estimated standard errors). Based on acceptable overall fit from all three models (LC, LCRE, and FM), in this case, researchers might conclude that the LCM is appropriate but that no reliable inference is available for item sensitivities.

For the dataset from Uganda, the estimated parameters under the LC, LCRE and FM models as well as goodness-of-fit statistics are provided in Tables 3 and 4 and also in the Shiny application under the Uganda example tab (https://perma.cc/QJ4V-GQ84). The models again perform similarly between them in terms of parameter estimates, however, the overall fit for this data set is poor across all three models (the overall $X^2$ ranges from 36.93, d.f. = 20 for LC to 40.25, d.f. = 18 for LCRE, to 44.42, d.f. = 19 for FM). Note that the overall significant (not an adequate model fit) result we obtained from the global goodness-of-fit tests cannot be trusted

**Table 1. Parameter estimates for the three model fits for the Ghana data set.**

|  | LC | LCRE | FM |
|---|---|---|---|
| Prevalence | 0.16 (0.04) | 0.15 (0.04) | 0.15 (0.04) |
| *Sensitivities* | | | |
| ELISA | 0.95 (0.04) | 0.97 (0.04) | 0.94 (0.05) |
| Ultrasound | 0.63 (0.10) | 0.66 (0.11) | 0.64 (0.10) |
| Haematuria | 0.74 (0.10) | 0.76 (0.11) | 0.78 (0.12) |
| Eggs | 0.84 (0.11) | 0.89 (0.13) | 0.84 (0.12) |
| Circ. Antigens | 0.48 (0.09) | 0.48 (0.10) | 0.46 (0.10) |
| *Specificities* | | | |
| ELISA | 0.37 (0.04) | 0.36 (0.04) | 0.37 (0.04) |
| Ultrasound | 0.74 (0.03) | 0.74 (0.03) | 0.73 (0.03) |
| Haematuria | 0.87 (0.03) | 0.87 (0.03) | 0.88 (0.03) |
| Eggs | 0.97 (0.02) | 0.96 (0.02) | 0.96 (0.02) |
| Circ. Antigens | 0.57 (0.04) | 0.57 (0.04) | 0.56 (0.04) |

The table shows the estimated prevalences, sensitivities and specificities with standard errors in parentheses for the Ghana data set.

**Table 2. Fitted frequencies for the three model fits for the Ghana data set.**

| | Pattern | Observed | LC | LCRE | FM |
|---|---|---|---|---|---|
| 1 | 00000 | 28 | 23.99 | 23.93 | 23.59 |
| 2 | 00001 | 19 | 18.48 | 18.28 | 18.53 |
| 3 | 00010 | 1 | 0.89 | 0.94 | 1.10 |
| 4 | 00100 | 4 | 3.49 | 3.58 | 3.43 |
| 5 | 00101 | 3 | 2.69 | 2.74 | 2.70 |
| 6 | 00110 | 1 | 0.34 | 0.25 | 0.42 |
| 7 | 01000 | 10 | 8.56 | 8.62 | 8.56 |
| 8 | 01001 | 2 | 6.60 | 6.59 | 6.73 |
| 9 | 01100 | 1 | 1.30 | 1.31 | 1.32 |
| 10 | 10000 | 34 | 41.53 | 41.88 | 41.08 |
| 11 | 10001 | 35 | 32.02 | 32.02 | 32.28 |
| 12 | 10010 | 2 | 2.79 | 2.74 | 2.87 |
| 13 | 10011 | 3 | 2.35 | 2.30 | 2.33 |
| 14 | 10100 | 6 | 6.68 | 6.68 | 6.61 |
| 15 | 10101 | 7 | 5.25 | 5.19 | 5.24 |
| 16 | 10110 | 5 | 4.11 | 3.98 | 4.14 |
| 17 | 10111 | 2 | 3.72 | 3.69 | 3.54 |
| 18 | 11000 | 19 | 15.16 | 15.30 | 15.18 |
| 19 | 11001 | 14 | 11.74 | 11.74 | 11.95 |
| 20 | 11010 | 2 | 2.90 | 2.81 | 2.60 |
| 21 | 11011 | 3 | 2.58 | 2.54 | 2.18 |
| 22 | 11100 | 3 | 3.41 | 3.12 | 3.46 |
| 23 | 11101 | 1 | 2.81 | 2.53 | 2.81 |
| 24 | 11110 | 7 | 6.81 | 7.26 | 7.09 |
| 25 | 11111 | 8 | 6.21 | 6.80 | 6.08 |
| | | $X^2$ | 12.79 | 13.23 | 12.42 |

The table shows the observed and expected frequencies under the three model fits for the Ghana data set. Note that the expected frequencies for the patterns 00000 and 11111 under the FM model fit are based on the non-deterministic component of the model. Adding these frequencies to the deterministic part gives us the total observed frequencies in these all zero and all one responses.

due to sparseness (i.e. there are response patterns with expected frequencies under the model less than 5). More specifically, the GF-Fit values are given in Table 1 in the S1 Appendix for the pair POC-CCA and SEA-ELISA diagnostic tests and for the LC, LCRE and FM models. The LC model gives a total GF-Fit equal to 10.37 (similarly the GF-fit values are 9.54 and 4.00 for the LCRE and FM models respectively). This is the total of the GF-fit values from the four cells. If we apply the rule of thumb given above, then any cell with a value greater than 16 indicates a bad fit or in other words the average across the four cells should not be greater than four. According to the rule of thumb, all three models show a good fit on the bivariate tables. On the evidence from the margins, we have no reason to reject any of the three models. The univariate and bivariate GF-Fits for the LC, LCRE and FM models are given in Table 2 in S1 Appendix for all pairs of diagnostic tests. Among the three models the LCRE provides the smallest bivariate GF-Fits but all models show adequate fit. Each value that appears in the table is smaller than 16.

Furthermore, for this same data set and the three examined models the empirical $p$-values are: LC: $X^2 = 43.51$, $p = 0.125$; LCRE: $X^2 = 47.032$, $p = 0.104$; FM: $X^2 = 50.6$, $p = 0.060$. These

**Table 3. Parameter estimates for the three model fits for the Uganda data set.**

|  | LC | LCRE | FM |
|---|---|---|---|
| Prevalence | 0.47 (0.03) | 0.48 (0.03) | 0.52 (0.05) |
| *Sensitivities* |  |  |  |
| POC-CCA | 0.98 (0.01) | 0.98 (0.01) | 0.98 (0.02) |
| SEA-ELISA | 0.98 (0.01) | 0.98 (0.01) | 0.98 (0.02) |
| DNA-TaqMan | 0.92 (0.03) | 0.92 (0.03) | 0.88 (0.04) |
| Kato-Katz 1 | 0.79 (0.04) | 0.79 (0.04) | 0.73 (0.06) |
| Kato-Katz 2 | 0.79 (0.04) | 0.79 (0.04) | 0.72 (0.06) |
| *Specificities* |  |  |  |
| POC-CCA | 0.80 (0.04) | 0.81 (0.04) | 0.88 (0.06) |
| SEA-ELISA | 0.46 (0.04) | 0.46 (0.04) | 0.51 (0.05) |
| DNA-TaqMan | 0.55 (0.04) | 0.55 (0.05) | 0.56 (0.05) |
| Kato-Katz 1 | 0.99 (0.01) | 1.00 (0.01) | 0.99 (0.01) |
| Kato-Katz 2 | 0.99 (0.01) | 1.00 (0.01) | 0.99 (0.01) |

The table shows the estimated prevalences, sensitivities and specificities with standard errors in parentheses for the Uganda data set.

values now suggest that all three models are a good fit to the data. This result is in line with the fact that the bivariate margins did not detect any misfits in the bivariate residuals.

In the absence of gold standard and solid prior knowledge on latent dependencies, can researchers make conclusions by using these results? In this paper, we attempt to address this broad question with the simulation study below.

**Table 4. Fitted frequencies for the three model fits for the Uganda data set.**

|  | Pattern | Observed | LC | LCRE | FM |
|---|---|---|---|---|---|
| 1 | 00000 | 39 | 26.81 | 27.48 | 29.89 |
| 2 | 00100 | 20 | 22.11 | 22.42 | 23.76 |
| 3 | 01000 | 22 | 31.48 | 32.22 | 29.06 |
| 4 | 01011 | 1 | 0.14 | 0.12 | 0.22 |
| 5 | 01100 | 28 | 26.05 | 26.38 | 23.29 |
| 6 | 01101 | 1 | 0.7 | 0.43 | 0.88 |
| 7 | 10000 | 1 | 6.62 | 6.46 | 4.08 |
| 8 | 10100 | 2 | 5.55 | 5.36 | 3.43 |
| 9 | 10110 | 2 | 0.45 | 0.38 | 0.61 |
| 10 | 11000 | 13 | 8.18 | 7.97 | 5.1 |
| 11 | 11001 | 2 | 1.72 | 1.59 | 3.13 |
| 12 | 11010 | 1 | 1.72 | 1.59 | 3.16 |
| 13 | 11011 | 5 | 6.2 | 5.98 | 8.16 |
| 14 | 11100 | 18 | 11.04 | 10.9 | 11.66 |
| 15 | 11101 | 14 | 17.69 | 17.97 | 22.42 |
| 16 | 11110 | 14 | 17.83 | 18.05 | 22.65 |
| 17 | 11111 | 75 | 67.27 | 68.45 | 59.31 |
|  | $X^2$ | | 36.93 | 40.25 | 44.42 |

The table shows the observed and expected frequencies under the three model fits for the Uganda data set. As in Table 2, the expected frequencies under the FM model fit for the all zero and all one responses only show the non-deterministic component.

## Design of the simulations

To resemble real world problems in the diagnosis of schistosomiasis, we consider five imperfect test items and the gold standard which we assume has 100% sensitivity and 100% specificity. In the simulation study, we consider changing four settings: the sample size, the disease prevalence, the availability of gold standard, and the data generating model. Thus, we specify two sample sizes: 1) 250, which is similar to a typical sample size in practice, and 2) 1000, which we use to demonstrate potential improvements solely due to a larger sample. We specify a high, 40% as an in-between value for *S. mansoni* and *S. haematobium*, and a low, 8%, disease prevalence settings to examine how latent variable modelling results might be affected by different prevalence levels. We consider cases when gold standard is not collected (100% of individuals are missing the gold standard), when gold standard is available for 20% (80% missing) and when gold standard is available for 50% (50% missing). Finally, we consider three latent variable models for generating the data: latent class (LC), latent class with individual-specific random effects (LCRE) and the finite mixture model (FM). These settings give us $3 \times 2 \times 2 \times 3 = 36$ simulation scenarios, and we use 128 replications for each one.

Because, in real world problems, we do not know the data generating process, we fit all three models for every simulated data set and study bias and mean squared error for the estimates of prevalence, specificities and sensitivities under the three hypothesised models. As in the real data example above, the models were fitted fully Bayesian with uninformative priors on the model parameters. The aims of the simulation study are two-fold. First, we explore the idea of sensitivity analysis by using latent structure models with different specifications of latent dependencies, similarly to Albert & Dodd [50] and Erosheva & Joutard [49] (but excluding the Grade of Membership and Extended Mixture Grade of Membership models in the current study), now with different settings of prevalence, sensitivity and specificity parameters that are particularly relevant for *S. mansoni*. Given that we do not know the data generating model, we examine whether one can make reliable conclusions about prevalence, sensitivity, and specificity for the given tests by relying on results from the various models and goodness of fit test. Second, because many empirical studies only use the LC, we focus on its performance in our simulations. We investigate how the standard LC model performs under CI violation in small and medium sample sizes, two differing proportions of prevalence, and three levels of available gold standard.

The simulated data have been generated using the sensitivities and specificities given in Table 5. Note that, for test items other than gold standard, the true sensitivity values range from 0.6 to 0.95 and the true specificity values range from 0.45 to 0.99. These scenarios were assumed to represent a real world problem for the diagnosis of *S. mansoni* infection.

**Table 5. Simulated parameter values for sensitivity and specificity for the diagnosis of *S. mansoni* infection.**

|  | Sensitivity | Specificity |
|---|---|---|
| Microscopy [54] | 0.60 | 0.99 |
| CCA [7] | 0.73 | 0.45 |
| CAA [55] | 0.90 | 0.87 |
| Antibody | 0.90 | 0.50 |
| LAMP | 0.95 | 0.90 |
| Gold std. [7, 8, 33] | 1.00 | 1.00 |

These parameter values are a mixture of best guesses of schistosomiasis diagnostics expert Prof. Clive Shiff and available published evidence. The relevant references are provided in the above table.

Following Erosheva & Joutard [49], we set $\sigma_0 = \sigma_1 = 1.5$ for the random effects variance in the LCRE model. The correlation between items will be attributed to two components, namely the latent class and the random effect. Given a latent class, the diagnostic tests will still be correlated based on the random effect and the higher the variance of the random effect the higher the correlation between the items from the same class. For generating the zero- and one-excess in the FM model, we set $\eta_0 = P(t = 0) = 0.5$ and $\eta_1 = P(t = 1) = 0.2$ for generating the zero- and one-excess in the FM model.

We use as performance criteria the bias and mean squared error (MSE) for each parameter given by: $Bias = \frac{1}{T}\sum_{i=1}^{T}(\hat{\theta}_i - \theta)$ and $MSE = \frac{1}{T}\sum_{i=1}^{T}(\hat{\theta}_i - \theta)^2$, where $T$ here is the number of valid replicates, $\hat{\theta}_i$ is the estimate of a parameter or of its asymptotic standard error at the $i$th valid replication, and $\theta$ is the corresponding true value.

## Simulation results

Simulation results show that the low prevalence setting could result in very unreliable estimates for prevalence and sensitivities for low sample sizes and in the absence of full gold standard. Moreover, even for larger sample size and particularly for low prevalence levels, not having any gold standard observations in the data could prohibit parameter recovery. We note that specificity parameters are generally estimated more reliably than sensitivity parameters in all scenarios (see Fig 1), consistent with findings from prior research [49]. Also consistent with prior research [49], we find when specificity parameters are biased, they are biased upwards. However, we do not observe a general trend in the direction of biases for sensitivity parameters. The sensitivity parameter estimates acquire large expected biases when the data generating model is LCRE (see Fig 1C); the biases are smaller but substantial when FM is the data generating model (see Fig 1E), and the biases are the smallest when LC is the data generating model (see Fig 1A). A similar pattern arises for the MSE of parameter estimates (see Fig 2). We also observe that it is difficult to distinguish between various forms of latent dependencies solely based on the goodness of fit results (see 'Model Fit' tab in https://perma.cc/QJ4V-GQ84), which is consistent with prior findings [49, 50].

Within each generating model scenario, having a large quantity of gold standard observations is improving the bias in estimates substantially. Thus, when gold standard is available for 50% of observations, we can conclude that specificity parameter estimates acquire little bias for either the large or the small sample size, irrespective of the data generating model and the two different assumed prevalence levels, and, most importantly, irrespective of the fitted model. Likewise, having 20% of gold standard improves with expected biases in sensitivity estimates, however, we still observe bias up to 0.1 for sensitivity estimates when the sample size is small and LCRE is the data-generating model. Notably, these biases are observed when a simpler LC model is fitted to LCRE-generated data, however, these biases are comparable to expected biases when LCRE model is fit to LCRE-generated data. Thus, sensitivity estimates from both models–the true LCRE and the simpler LCM–acquire expected biases in this case. As expected, these biases improve and become potentially negligible for the larger sample size of 1000. The simulated results under the LC, LCRE and FM models data generating mechanisms as well as for the two different assumed sample sizes and prevalences are provided in the Shiny application (https://perma.cc/QJ4V-GQ84) under the 'Simulated results' tab and by selecting corresponding different simulation scenarios within this tab. Interestingly, the expected impact on parameter estimates due to using an incorrect fitting model is quite negligible as compared to the impact on parameter estimates due to the absence of gold standard particularly for the sensitivity estimates.

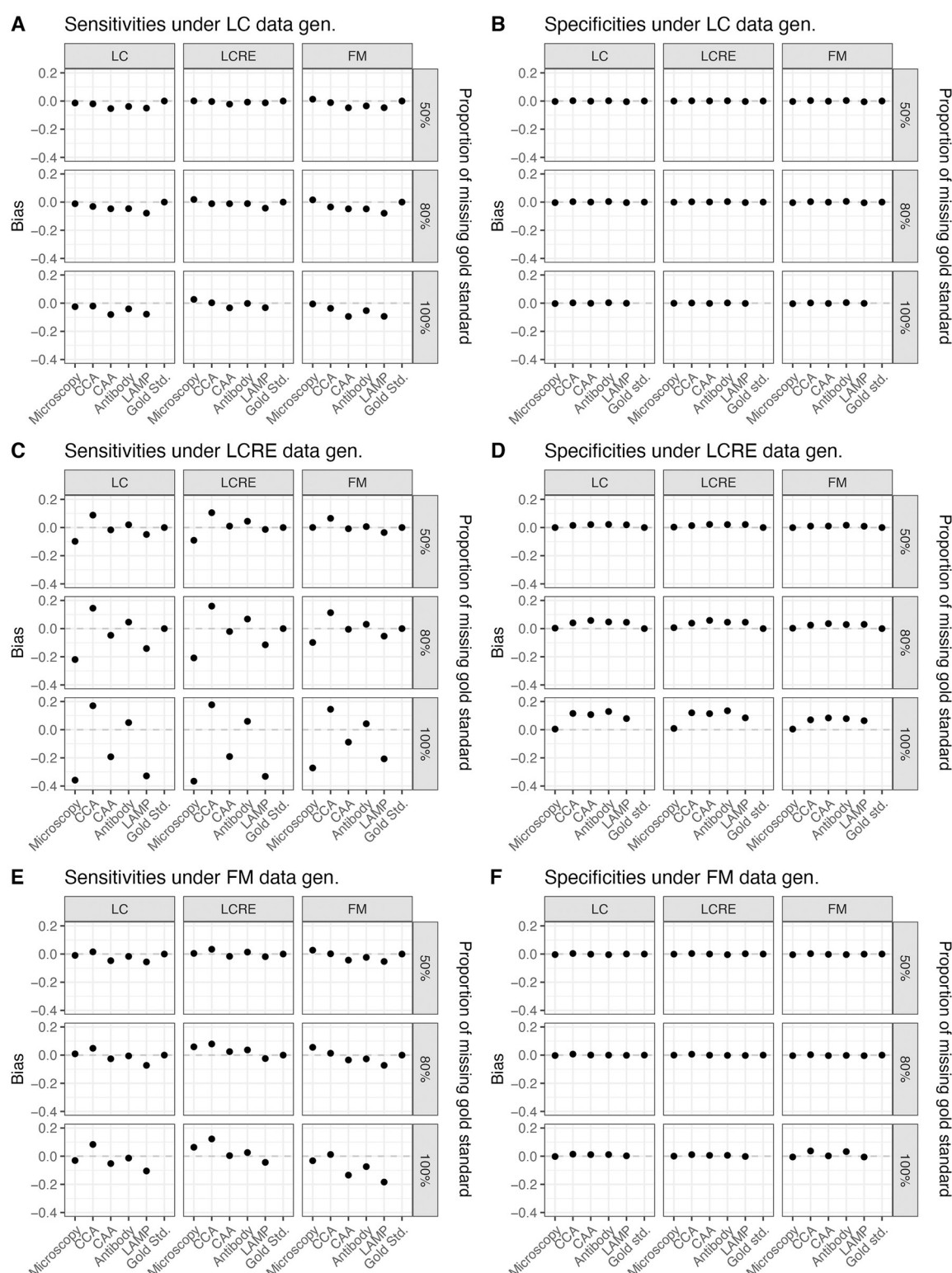

**Fig 1. Bias of parameter estimates (sensitivities and specificities), as estimated by the LC, LCRE and FM models under differing proportions of missing gold standard, and under differing data generating mechanisms (sample size = 250, prevalence = 0.08).**

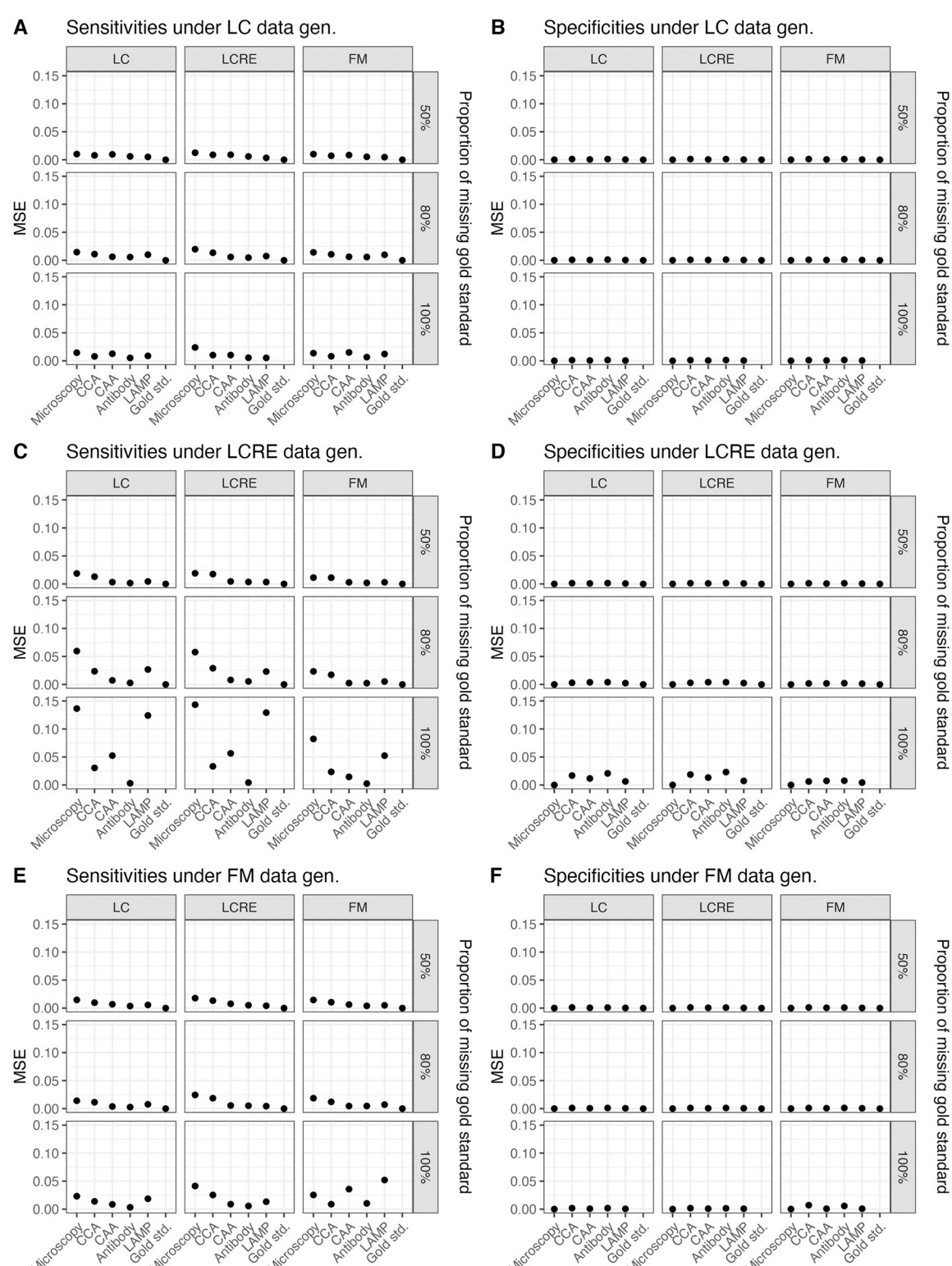

**Fig 2. MSE of parameter estimates (sensitivities and specificities) as estimated by the LC, LCRE and FM models under differing proportions of missing gold standard, and under differing data generating mechanisms (sample size = 250, prevalence = 0.08).**

We observe that parameter recovery is less reliable when latent dependency structures are more complex than under the CI-LCM, with LCRE data-generating scenario being the most problematic. Thus, for the small sample size of 250, and 8% prevalence, when no gold standard is available and data are generated from the LCRE model, we find that the expected bias in sensitivity estimates ranges from −0.4 to 0.2, with high specificity items experiencing negative biases that are largest in magnitude. Results improve in the case of 40% prevalence and sample size of 250 (see S1 and S2 Figs). Under the LCRE generating model, inspecting the fit of the three models, we find that this cannot be considered adequate (even when taking into account that the data are quite sparse); however, we find that estimates from different models are in agreement. In this setting, using the LCRE model for prevalence estimation could result in erroneously estimating the prevalence to be much higher than it actually is (e.g., this could range from 12.3% at the 80th replication to 37.5% at the 100th replication versus the assumed 8%). For one to obtain the different model estimates and fit across the 128 replications, in the Shiny app go to tab 'Model fit' and type successively 1–128 in the box 'Replication number'). When LC model was the data generating model, all three models showed good model fit and reasonable estimates of prevalence. When FM was the data generating model, the prevalence estimates were also reasonable for the bigger sample size.

To conclude our observations from the simulation studies, we recommend that in the absence of strong scientific knowledge about the data generating mechanisms, one needs to be careful when interpreting prevalence and sensitivity estimates from latent structure models in the absence of gold standard especially in small sample sizes and in the presence of low prevalence. The simulation results indicate that the availability of LCMs and their extensions does not automatically imply that assessing the true disease status is unnecessary. In fact, under realistic scenarios of a few test items and a sample size of a couple hundred observations, it is possible to obtain severely biased estimates, especially for prevalence and sensitivity parameters, when the true data generating mechanism is more complex than standard LCM with CI. The biases should be of utmost concern when true disease prevalence is low. Based on the simulation studies, we recommend to carefully examine model fit before drawing any conclusions. In our simulations, the fit of LC, LCRE and FM models was poor across the board for small sample sizes when data generating models were complex. If the fit was found to be poor, in the case of small prevalence, sensitivity and prevalence estimates were found to be not reliable even if there is an agreement among different latent structure models. In such cases, we recommend collecting some gold standard. Our simulation results indicate that even 20% of gold standard can drastically improve estimation. Another alternative, which we did not explore in this paper, is in the absence of gold standard to consider covariates such as gender and age on the prevalence and item response probabilities (i.e. the sensitivities and specificities of the diagnostic tests) and/or Bayesian estimation with informative priors.

## Discussion

In schistosomiasis diagnostic studies, several inherent study design issues might compromise accurate estimation of parameters from latent variable models. More specifically, those study design issues are: the inclusion of small number of diagnostic tests (i.e. most often ≤5) with different mutual dependencies in diseased and non-diseased subjects; the lack of a gold standard due to mainly lack of appropriate equipment and training of technicians across the endemic countries because of scarce financial resources; the current use of relatively small sample sizes in relevant studies due to lack of research funding, and finally the inherent differing levels of prevalence of the studied disease even within the same area of one endemic country. In general, some of these issues can also be pertinent in the diagnosis of other major

tropical infectious diseases such as malaria, tuberculosis, dengue and soil transmitted helminths and thus we hope that current findings would be also considered and adapted more broadly.

We have initially fitted three different latent variable models–LCM with CI assumption, LCRE and FM–in two real schistosomiasis datasets from sub-Saharan Africa of small sample size and with a limited number of diagnostic tests in order to caution the practitioners not to blindly apply such methods for estimating diagnostic error without a gold standard. We have subsequently explored solutions from the biostatistical literature [50] of whether including a partial gold standard in these models improves bias and precision of parameter estimates for the problem of schistosomiasis diagnosis via extensive simulation studies (assuming different sensitivities and specificities for different diagnostic tests and a low- and a high-prevalence settings). All of these are data-driven modelling approaches searching for relationships in the available data and accounting for measurement error and dependencies between the diagnostic tests as well as excessive number of zero and one responses, aiming for accurate parameter inference. The latter is a key component for correct predictions from mechanistic approaches, which incorporate the available knowledge of the system into the model [56]. Combination of data-driven and mathematical modelling approaches could offer great synergies with regards to the understanding of interactions between population epidemiology and diagnostic tests characteristics. Our simulations did not take into account that the Kato Katz sensitivity might change with infection intensity as shown in [16, 57, 58], and this can be an interesting topic for further research to refine the current simulations. Another interesting topic for further research can involve another set of simulations in which the gold standard is not allowed to have 100% sensitivity and specificity but close to them as a means to examine additional biases of assuming a diagnostic as a gold standard when it is not. Nevertheless, the current simulation results still provide useful recommendations for researchers to consider in their study design and analysis.

For the real dataset from Ghana, all three models provided acceptable overall model fit, with all the bivariate chi-squared residuals also indicating a good fit. Estimates of prevalence, sensitivity and specificity were all similar across the three fitting models and thus in this case, researchers might conclude that the LCM is appropriate but that no reliable inference is available for item sensitivities (due to large standard errors) in the absence of partial gold standard information. However, for the real dataset from Uganda although the models performed similarly in terms of parameter estimates, the overall fit was poor across all three models but with all the bivariate $\chi^2$ residuals indicating a good fit. For the specific dataset from Uganda and based on our simulation results, we conclude that inclusion of partial gold standard information would have added more confidence in the parameter estimates and derived conclusions. It should be also noted that if the inclusion of partial gold standard is not feasible, then alternatively the inclusion of covariates or Bayesian estimation with informative priors [58] might have improved model fit.

Furthermore, our simulation studies indicated that the sensitivity parameter estimates acquired some degree of expected biases in small sample sizes. We explain the reasons for these findings below. For instance, in our simulation scenarios, when we generate data from the FM with an excess for all-one and all-zero responses and a sample size of 250 we are left with 75 observations to be fitted by the hypothesised model. When prevalence is also low, we then get very unstable results. On the other hand, the LCRE fits a continuous latent variable (random effect) in addition to the two latent classes to explain the dependencies among the five (in the full absence of gold standard) or six items (when there is partial gold standard information), respectively, beyond the ones explained by the two latent classes. With an LCRE we do not estimate loadings for the random effects but again the model is probably overfitting

and could also result in unstable estimation. Another inherent problem with the LCRE fitting for the scenario of the small sample size, is the difficulty of estimating random effects variance from small samples. Those two might be the reasons that even when we estimate from LCRE or FM models to their respective data generating models, the number of items is small or the sample size is small after we eliminate the 1 and 0 response patterns.

More generally, the simulation studies showed that the fit of all three examined models was poor in the case of low prevalence and that partial gold standard information improved the accuracy and bias of parameter estimates (prevalence, sensitivities, and specificities for the examined diagnostic tests) in the presence of model violations. In particular, when a future study could afford recruiting only around 250 participants, we recommend the inclusion of at least 20% gold standard information (i.e., collect gold standard on 50 randomly selected individuals of the initial sample) and, most ideally, of 50% gold standard (i.e., collect gold standard on 125 randomly selected individuals of the initial sample) for both low and higher prevalence scenarios. However, even for sample size of 1000, particularly for low prevalence levels, the full absence of gold standard could lead to erroneous parameter estimates but inclusion of 20% gold standard (i.e., gold standard collected on 200 randomly selected individuals of the initial sample) can also notably improve the obtained parameter estimates. If a high prevalence is expected, and the sample size can be as large as 1000, the inclusion of partial gold standard for the improvement of bias and precision of model estimates becomes less important.

For all different prevalence levels and sample sizes, we definitely recommend to carefully examine different models fit of real world data before drawing any conclusions. Latent class models can be fitted to binary but also other type of indicators such as continuous, ordinal or mixed. Sensitivities and specificities would need to be carefully considered for those type of data too but in principle the examined models of the current study can be extended to handle any type of data (such as original egg counts and CCA results) [59, 60]. It should be noted that the more complex models presented here are not possible to be fitted with less than five diagnostic items. With a higher number of items, it could also be warranted to explore other formulations for latent dependencies such as mixed membership [61], which may not necessarily be helpful for only five items as in our case because of known difficulties in determining true latent dependency structures [49, 50]. The openly available datasets in from the paper by Bärenbold et al. at [58] with up to 12 diagnostic tests from many different sites and in total several thousand individuals or the dataset used in the paper by Prada et al. in 'additional file 3' [57] can be good sources to try and apply such models. In applied studies where only three or four diagnostic tests would be available and no gold standard would be feasible to be collected, we would also recommend Bayesian inference and the inclusion of informative priors on latent class modelling parameters. Such an approach would represent our (un)certainty in the model parameters and this could also improve the model estimation accuracy [62, 63]. The inclusion of informative priors and their impact on certain parameters compared to the impact of adding gold-standard information is definitely a subject that warrants further study but this is out of scope of this article.

Finally, future methodological research should also explore the more precise merits of incorporating covariates (such as gender and age) on the prevalence and item response probabilities and how those could improve the fit of these models. Overall, the inclusion of covariates can strengthen the prediction power of a model. With latent variable models such as LC, LCRE and FM, the inclusion of individual level characteristics on the sensitivities and specificities can identify individual-item effects that might be of medical importance and interest (for e.g., diagnostic tests behaving differently between men and women or different age groups) but also covariates can be seen as variables that together with the latent structure explain local dependencies and reduce the variance of the random effects. The inclusion of individual level

covariates on the prevalences can identify groups of individuals that are more or less prone to a disease. For instance, in schistosomiasis, chronic infections occur in adults of increasing age, but in these groups it has been difficult to detect infection based on egg detection while serological and immunological tests can be more appropriate to detect duration of infection [64] so assessing diagnostic accuracy for different age groups in schistosomiasis studies, is highly relevant. In addition, gender may influence the accuracy of estimates through factors such as menstruation and genitourinary tract infection in females [65] and thus gender is another highly relevant covariate to be considered. Such approaches have been also empirically explored and applied in schistosomiasis diagnostic accuracy studies [8]. Other covariates relevant to diagnostic accuracy could be multiple labs or technicians, different locations, schools or different countries. Such scenarios could be possible if identical training or equipment is not used across the different locations, or if multi-country diagnostic studies were in place and the researcher analysing the data would be keen to check and estimate such differences.

There are currently various debates and initiatives within the global health arena towards the elimination of schistosomiasis within the next decade [66]. Thus, operational research accurately evaluating existing and new diagnostic tools as well as quantifying the epidemic status for guiding effective and well-focused strategies is essential [67]. Our article apart from outlining the mathematical details of these models and their optimal usage for modelling diagnostic errors in the context of schistosomiasis, also provides the JAGS code so that readers can fit the discussed models to other relevant datasets and perform their own sensitivity analysis. In this way, we strongly believe that the current article could contribute notably valuable tools to the operational research agenda mentioned above.

## Supporting information

**S1 Appendix. Results (fit measures) for the Uganda data set.** Univariate and bivariate GF-Fits for the LC, LCRE and FM models for the Uganda data set.
(PDF)

**S1 Fig. Additional bias plots from simulation study.** Bias of parameter estimates (sensitivities and specificities) as estimated by the LC, LCRE and FM models under differing proportions of missing gold standard, and under differing data generating mechanisms (sample size = 250, prevalence = 0.4).
(TIF)

**S2 Fig. Additional MSE plots from simulation study.** Mean squared error of parameter estimates (sensitivities and specificities) as estimated by the LC, LCRE and FM models under differing proportions of missing gold standard, and under differing data generating mechanisms (sample size = 250, prevalence = 0.4).
(TIF)

**S1 File. JAGS code.** JAGS code for fitting the latent class model under conditional independence, the latent class model with Gaussian random effects, and the finite mixture latent class model.
(PDF)

## Acknowledgments

The authors deeply acknowledge the support of Professor Russell Stothard for provision of the Ugandan dataset and useful scientific comments and advice throughout this article.

## Author Contributions

**Conceptualization:** Artemis Koukounari.

**Data curation:** Artemis Koukounari.

**Formal analysis:** Haziq Jamil.

**Investigation:** Artemis Koukounari, Haziq Jamil, Elena Erosheva, Clive Shiff, Irini Moustaki.

**Methodology:** Artemis Koukounari, Haziq Jamil, Elena Erosheva, Irini Moustaki.

**Project administration:** Artemis Koukounari, Irini Moustaki.

**Resources:** Clive Shiff, Irini Moustaki.

**Software:** Haziq Jamil.

**Supervision:** Elena Erosheva, Clive Shiff, Irini Moustaki.

**Validation:** Haziq Jamil.

**Visualization:** Haziq Jamil.

**Writing – original draft:** Artemis Koukounari.

**Writing – review & editing:** Haziq Jamil, Elena Erosheva, Clive Shiff, Irini Moustaki.

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
