## [Decision Letter · Decision Letter 0]

11 Jun 2020

Dear Dr Koukounari,

Thank you very much for submitting your manuscript "Modeling and design issues for schistosomiasis diagnostic studies" for consideration at PLOS Neglected Tropical Diseases. As with all papers reviewed by the journal, your manuscript was reviewed by members of the editorial board and by several independent reviewers. In light of the reviews (below this email), we would like to invite the resubmission of a significantly-revised version that takes into account the reviewers' comments. 

Two of the reviewers highlight some significant issues with the manuscript, which I would like to see addressed. In particular, in the known absence of a gold standard for schistosomiasis, and even a partial gold standard as the paper refers to, it is difficult to improve studies looking at different diagnostic methods. This paper highlights these issues but does not really provide the public health community or Schistosoma research community with a viable solution. The paper would be greatly improved and suitable for resubmission for review at PLoS NTDs if as Reviewer 1 suggests it presented what might be the least worst alternative study design for diagnostics analysis, including study size, prevalence, and number of tests compared. In addition the inclusion of additional published or simulated datasets, as suggested by Reviewer 3 could also improve the robustness of any conclusions.

We cannot make any decision about publication until we have seen the revised manuscript and your response to the reviewers' comments. Your revised manuscript is also likely to be sent to reviewers for further evaluation.

Sincerely,

Poppy H L Lamberton

Guest Editor

Maria Elena Bottazzi

Deputy Editor

Two of the reviewers highlight some significant issues with the manuscript, which I would like to see addressed. In particular, in the known absence of a gold standard for schistosomiasis, and even a partial gold standard as the paper refers to, it is difficult to improve studies looking at different diagnostic methods. This paper highlights these issues but does not really provide the public health community or Schistosoma research community with a viable solution. The paper would be greatly improved and suitable for resubmission for review at PLoS NTDs if as Reviewer 1 suggests it presented what might be the least worst alternative study design for diagnostics analysis, including study size, prevalence, and number of tests compared. In addition the inclusion of additional published or simulated datasets, as suggested by Reviewer 3 could also improve the robustness of any conclusions.

Reviewer's Responses to Questions

**Key Review Criteria Required for Acceptance?**

**Methods**

-Are the objectives of the study clearly articulated with a clear testable hypothesis stated?

-Is the study design appropriate to address the stated objectives?

-Is the population clearly described and appropriate for the hypothesis being tested?

-Is the sample size sufficient to ensure adequate power to address the hypothesis being tested?

-Were correct statistical analysis used to support conclusions?

-Are there concerns about ethical or regulatory requirements being met?

Reviewer #1: (No Response)

Reviewer #2: The authors proposed to asses the fit of three different types of model to real data - latent class models with the conditional independence assumption met, a random effect latent class model and a finite mixture model, and to use simulated data to explore how improved gold standard measures would affect sensitivity and specificity estimates. Overall the work seems to be very thorough and well thought out. It address an interesting question in a time when epidemiological modelling comes under considerable scrutiny. 

There is no comment on the ethics of the data used. Indeed there is no real methods section that covers the data at all. I think this is quite bold given that the authors state lacking or questionable data as a problem for LCA. It also makes following the manuscript quite difficult. 

Essentially this paper could provide a valuable toolkit for those with less expertise but wishing to use the correct modelling method. However, the methods are diffuse throughout the paper, making it difficult to follow and hard for these methods to be extracted and replicated (I appreciate that the scripts are available but the text has to complement this). I would recommend making a very clear cut methods section first, that covers the data (real and simulated) and the models and the model selection all in one section. I would also recommend some sort of table or schematic (or combination) that lays out the differences in the three models and the differences in the goodness-of-fit tests. 

On that note, it is difficult to ascertain whether the authors are recommending against the first GOF test due to sparseness, or rather that those using these models should use both GOF tests. Again I think this could be cleared up with more definitive explanations here. 

From the introduction to the 'results with real data' section is quite long winded and repetitive. By streamlining this the authors would be able to really put a lot more detail into the methods, which seeing as this comes across first and foremost as a methods paper, would make sense. 

Small methodology side notes:

- I am unsure that 40% prevalence constitutes a truly high prevalence setting in the simulations. Perhaps this is an in-between value for S. mansoni and S. haematobium, but could probably have a small line of explanation. 

- In line 209 the authors state that there are 5 binary variables for both data sets, but the data for S. mansoni has only 4 diagnostic tests so what is the last variable. Again this could be cleared up with some sort of schematic or table, even in the SM.

Reviewer #3: -

**Results**

-Does the analysis presented match the analysis plan?

-Are the results clearly and completely presented?

-Are the figures (Tables, Images) of sufficient quality for clarity?

Reviewer #1: Table 1 is unclear. The authors state that these are tests for Schistosoma mansoni infections. However, it is not clear how dipsticks are relevant. Typically, any dipstick for schistosome infection diagnosis tests for microhematuria, which is only present in persons with Schistosoma haematobium infections. If there is another dipstick for S. mansoni, it should be described. Also, it is not at all clear how they determined the sensitivities and specificities for the different tests--are they hypothetical, best guesses, or available in published literature? Obviously, published literature is best but it must be cited. In addition, as stated elsewhere, the gold standard test for schistosomiasis diagnosis is a mythical beast.

Reviewer #2: I do not think the results section delivers the results in a clear, concise manner. There are a lot of methods and discussion dotted through it. For the reader to really make the most of the results this really needs streamlining. 

Perhaps tables comparing outputs of the models would be helpful, particularly in the real data results section as there is no mention of any figures within the real data results section for readers to refer to. The Shiny App here is helpful, but something visual and immediately available for readers to look at whilst reading would be very helpful. 

During the simulations results, there is a consistent message that more gold standard observations improve the model outputs in various ways. However, at no point is there a mention of the fact that there isn't a schistosomiasis gold standard. So what this 100% specificity 100% sensitivity gold standard observation is based on in reality, I can not see.

Reviewer #3: The estimated sensitivity of 80% of a single Kato-Katz in the Uganda dataset is inconsistent with the literature while the specificity of POC-CCA is most likely underestimated with 80%. The sensitivity of Kato-Katz is often determined in comparison with multiple Kato-Katz test as the 'gold' standard which has a very high specificity but imperfect sensitivity. Therefore, the sensitivity of a single Kato-Katz test, which generally is in the order of 50% in these studies, is overestimated and can be used as an upper bound. The implausible estimates of the sensitivity of Kato-Katz is indicative of the used models for the conditional dependence not fitting the data generating process well as explained also below.

**Conclusions**

-Are the conclusions supported by the data presented?

-Are the limitations of analysis clearly described?

-Do the authors discuss how these data can be helpful to advance our understanding of the topic under study?

-Is public health relevance addressed?

Reviewer #1: The conclusions that latent class modeling is not very accurate in the absence of a gold standard is well supported. However, because no gold standard exists, there is not a great deal of public health relevance or a pathway forward to advance understanding of schistosome diagnostic test evaluation.

Reviewer #2: The discussion is clear and well thought out. However, there are bits throughout the results that I think really belong in here. Though the implications for public health are referred to, I do think that overall the authors make many suggestions without any mention of their application in the field. I think that this cross-over is an important consideration in modelling studies and will add weight to the findings, making them more accessible to non-modelling readers. With that said, I am still not sure how the authors address the lack of a gold standard to begin with, if they suggest that this should be included in the model more often. I appreciate that this is somewhat dependent on the infection system in question, and varies between S. haem. and mansoni, but as these are the datasets the authors have used, it deserves a some discussion. 

There is also no real discussion of the other ways in which more recent modelling studies use LCA and do attempt to address the lack of a gold standard. I would like to see more discussion on how the three models here compare to the current literature.

**Summary and General Comments**

Reviewer #1: The authors' main conclusion, which is probably correct, is that the way latent class analyses have been done when comparing different tests for schistosome infections has severe limitations. However, their solution to this problem, to use a "gold standard" test to evaluate at least a portion of the samples, is in itself problematic. Namely, there is no gold standard test for schistosome infections. If there were a gold standard test, latent class modeling would be unnecessary--the alternative tests could just be compared to the gold standard to determine their sensitivity and specificity. They allude, very briefly, to a couple of molecular assays that "could" be a gold standard for schistosomiasis diagnostics but provide no evidence why these tests are any better than all of the other imperfect tests available. The claims that these tests could be gold standards are only backed up by the authors of the referenced papers.

Thus, while the authors point out what is likely a very real problem with the way analysis of diagnostics is currently done, they really provide no viable solution, making the paper not all that useful. What would be more useful is while acknowledging that no gold standard exists, presenting what might be the least worst alternative study design for diagnostics analysis, including study size, prevalence, and number of tests compared (although this is likely to be limited by what is available as well).

The authors should also present their findings in the context of the particular use case for the assay as this can greatly influence the test utility. For example, testing antibody levels can be useful for initial mapping where no previous treatment has been conducted or following presumed interruption of transmission in younger age groups. However, antibody testing is not useful at all to monitor and evaluate an ongoing treatment program in school children.

Reviewer #2: Overall: 

This paper addresses an important topic and could provide a really amazing toolbox. The author is clearly well informed and is passionate about the proper use of the tool and I can totally see why. I also think the Shiny App is a nice way to visualise what they have done, it really helped me to understand what the purpose of the paper was. 

With that said, there is so much information here, that is very diffuse in its presentation, particularly in the introduction. This makes it very difficult to find the purpose of the paper until much later in. The results and the Shiny App therefore don't stand out and the shiny App should not be used as a crutch for results to lean on. Indeed the results are hard to tease out, despite actually being very interesting and quite important to this field. 

My greatest suggestion would therefore be to restructure and streamline a lot of the paper. 

Some additional comments: 

Abstract:

When the authors mention issues of the model when applied to a small number of diagnostics, do this mean a small number of different types of diagnostics (i.e. KK vs, KK & CCA, vs KK, CCA & CAA) or a small number of results from each diagnostic (i.e. sample size)? This becomes clear a little later on, but I’d expand on this slightly here as it is confusing.

Author Summary: 

“the lack of a gold standard due to mainly lack of appropriate equipment and training of technicians across the endemic countries because of scarce financial resources” 

This is quite a narrow statement that entirely misses out the variability of certain methods due to biological phenomena that would occur even for the most highly trained technician with the most expensive equipment (i.e. KK egg variability, CCA detecting juveniles etc). It also does not account for the fact that counting worms in a host would be too invasive unless that host is dead. 

Main Body:

Lines 91-92: This would suggest quite a fundamental flaw in the data collection approach but unlike other paragraphs there is no citation to support that this has indeed happened anywhere. If the authors can find no evidence for it then say this is an additional consideration, or preferably expand this sentence into a paragraph and add an example. 

Line 76: Individuals do not have to be from low middle-income countries be co-infected. Surely individuals in any setting where both infections occur can experience co-infection - I understand they are connected but it is not a pre-requisite. 

Lines 94-121 are hard to follow. The authors give lots of examples and details of models and authors, but I do not think they are tied into a clear progression of the paper. Perhaps a table with the models and a summary of what they achieve or a streamlining of the section that feeds more concisely into what the authors propose to do in the final paragraph of this section, would be easier to follow. 

On from the previous comment, in the final line of the paragraph, it sounds like the authors will move into what they propose to do, however your next section brings up types of models mentioned in only parentheses at the top of the previous section. I suggest streamlining these two sections and making clearer the purpose of them in relation to the purpose of the paper. These models presented in the second paragraph are the crux of the work and should be mentioned more thoroughly earlier on. On that note, remove some of the more crucial model description from the SM and put in the main text to aid the reader.

Line 124: “Albert & Dodd proposed [insertion: a] solution” 

Lines 126 – 128: Albert & Dodd were talking about radiography and human error, rather than biological variability. It would be helpful here if the authors gave some examples of how this is possible in a setting where the gold standard cannot be acquired because it would be too invasive, or to have another approach for a gold standard that can be conducted in the field. I believe Lamberton et al (2014) would be a good source here. 

Line 150 – typo “Goodness-of-fit test statistics and measures of fit”

Lines 355-377: This really is discussion material.

Lines 275-276: I do not think the authors mention these parts of the model anywhere else in the paper. They do no not need to be mentioned here, it just adds confusion to readers who have otherwise never been introduced to these models.

Reviewer #3: The manuscript is a well written methods paper comparing three ways to incorporate conditional dependence between diagnostic tests using two datasets. It is evaluated whether including an additional gold standard diagnostic test for some of the individuals would improve overall estimation of diagnostic sensitivity and specificity. Results are presented in an innovative way using a shiny app.

I agree with the general message of the manuscript that having a good diagnostic test among the set of diagnostic tests improves estimation of sensitivity and specificity. However, I am not convinced of the usefulness of the results and used models for the specific case of schistosomiasis for the following reasons:

1. The authors discuss including gold standard diagnostic tests in parallel to improve estimation. However, the sources mentioned do not state that PCR can be considered such a gold (or even silver) standard for schistosomiasis. The source below for example estimates the sensitivity of PCR for schistosomiasis to 69% and the specificity to 91%. It is somewhat self-evident that including a diagnostic test with fully known sensitivity and specificity improves estimation in latent class models. Is the contribution of this paper mostly to quantify that?

Colley, Daniel G., et al. "A five-country evaluation of a point-of-care circulating cathodic antigen urine assay for the prevalence of Schistosoma mansoni." The American journal of tropical medicine and hygiene 88.3 (2013): 426-432.

2. The models models used to incorporate conditional dependence between diagnostic tests do not match the data generating process for schistosomiasis particularly well. For example the paper ignores the effect of infection intensity on diagnostic sensitivity (except one remark on line 85 as one effect among many) which is a if not the major driver of conditional dependence between different diagnostic tests. Schistosomiasis is not like many other conditions a binary status of either having the disease or not but diagnostic tests and morbidity depend on the number of worm-pairs harboured by an individual. Many studies have shown that the sensitivity changes strongly with infection intensity for example for both Kato-Katz and POC-CCA the relation is also given in the paper by Colley et al. in the reference above. There have been examples of applications of latent lass models with random effect structures inspired by what is known about the data generating process of schistososomiasis that are not at all discussed in this paper and that could be used to create more realistic simulations (see references below). Therefore, I doubt how relevant the simulation results are for the specific case of schistosomiasis. 

Prada, Joaquín M., et al. "Understanding the relationship between egg-and antigen-based diagnostics of Schistosoma mansoni infection pre-and post-treatment in Uganda." Parasites & vectors 11.1 (2018): 21.

Bärenbold, Oliver, et al. "Translating preventive chemotherapy prevalence thresholds for Schistosoma mansoni from the Kato-Katz technique into the point-of-care circulating cathodic antigen diagnostic test." PLoS neglected tropical diseases 12.12 (2018): e0006941.

3. There are only two small datasets used in this study to attempt to make a selection between the three proposed models. One with 220 and the other with 258 individuals and combined maybe about 160 positive individuals. I don't think that it is possible to select the most suitable model with that little data, especially when the models are fit to the two datasets separately. This could for example be tested by simulating datasets of different sizes from each model and checking how often the correct model will be selected with the used methodology. To increase power I suggest using for example the openly available data from the paper by Bärenbold et al. at https://doi.org/10.1371/journal.pntd.0006941.s002 with up to 12 diagnostic tests from many different sites and in total several thousand individuals or the dataset used in the paper by Prada et al. at in 'additional file 3' at https://doi.org/10.1186/s13071-017-2580-z.

PLOS authors have the option to publish the peer review history of their article (what does this mean?). If published, this will include your full peer review and any attached files.

Reviewer #1: No

Reviewer #2: No

Reviewer #3: No
---

## [Decision Letter · Decision Letter 1]

23 Oct 2020

Dear Dr Koukounari,

Thank you very much for submitting your manuscript "Modeling and design issues for schistosomiasis diagnostic studies" for consideration at PLOS Neglected Tropical Diseases. As with all papers reviewed by the journal, your manuscript was reviewed by members of the editorial board and by several independent reviewers. The reviewers appreciated the attention to an important topic. Based on the reviews, we are likely to accept this manuscript for publication, providing that you modify the manuscript according to the review recommendations. In particular, please could you 1) modify where you can without additional experiments being required, and 2) expand the 'Limitations' type paragraph in the discussion to be open about the concerns of the reviewer 3 especially with respect to gold standards or lack thereof.

Sincerely,

Poppy H L Lamberton

Deputy Editor

Maria Elena Bottazzi

Deputy Editor

Having read the submission again and the three reviewers comments, I am happy to go for a minor revision, but please could you 1) modify where you can without additional experiments being required, and 2) expand the 'Limitations' type paragraph in the discussion to be open about the concerns of the reviewer 3 particularly with respect to gold standards or lack thereof.

Reviewer's Responses to Questions

**Key Review Criteria Required for Acceptance?**

**Methods**

-Are the objectives of the study clearly articulated with a clear testable hypothesis stated?

-Is the study design appropriate to address the stated objectives?

-Is the population clearly described and appropriate for the hypothesis being tested?

-Is the sample size sufficient to ensure adequate power to address the hypothesis being tested?

-Were correct statistical analysis used to support conclusions?

-Are there concerns about ethical or regulatory requirements being met?

Reviewer #2: (No Response)

Reviewer #3: -Latent class Gaussian random effects model:

A gaussian random effect is included in each latent class to model dependence between different diagnostic tests. However, there is no diagnostic test specific parameter that determines the influence of the random effect on the sensitivity/specific. In particular I don't see how this model captures the situation where some diagnostic tests are correlated and some are not. This is not solved by the Finite mixture latent class model either.

-Goodness-of-fit test statistics measures of fit:

It is reported that the number of events in the two real datasets is too small for the Chi-squared test. This should indicate that more data is needed to evaluate whether any of the models sufficiently fit the latent correlation structure of the data. 

-Inclusion of partial gold standard:

Reference 5 only states that DNA detection is better than existing ones but not that it is a gold standard. Reference 6 is not about schistosomiasis in particular and does not report diagnostic accuracy. Reference 7 is a study in 89 people with Kato-Katz, CCA, and DNA detection which is certainly not sufficient to establish a gold standard. Reference 8 reports 100% sensitivity and specificity without confidence intervals from a dataset with urine filtration, dipstick, and DNA. It reports very low specificity for dipstick probably because all dipstick positive but DNA negative are treated as false-positives. I don't think this supports DNA as a gold standard with any confidence as I can't see how these estimates could be determined. Therefore, the current lack of gold standard used is not due to lack of equipment as stated in the author summary but according to current knowledge due to no gold standard existing. The authors argue that in absence of a gold standard the manuscript is still useful to discuss what models should be used and when to be cautious.

Reviewer #4: Please see general comments

**Results**

-Does the analysis presented match the analysis plan?

-Are the results clearly and completely presented?

-Are the figures (Tables, Images) of sufficient quality for clarity?

Reviewer #2: (No Response)

Reviewer #3: Results from real datasets:

-Table 1: Sensitivity of eggs is probably overestimated and specificity of ELISA and Circ. Antigens underestimated. I'm suprised that the LCRE model does not change estimates for antigen based tests which are probably heavily correlated but given the model allows only for dependence between all tests or none this might not be a surprise.

-Table 2: How come the FM models underestimates the 00000 and 11111? I thought point masses are included for those options and should therefore match the data?

-Table 3: A sensitivity of 79% of Kato-Katz and a specificity of 55% for a DNA based method are not realistic values.

-Table 4: The occurence of 00000 and 11111 are again understimated suggesting that the correlation structure is not captured by the model. I still don't understand how the FM model is not able to capture it.

-It is reported that for mansoni the fit is worse which could just reflect that the prevalence is higher and therefore there is more power to see discrepancies between the data and the models.

- what are the estimates for the parameters describing the structure violating the conditional independence in the more complex models?

-The results indicate that none of the models fits the data well enough but the sample size is too small to know. I'd be interested in a paper putting the models to a severe test to evaluate if any of them is useful to simulate data for schistosomiasis which is why I suggested using larger datasets.

Designs of the simulations:

- The sensitivity and specificity values chosen for the simulation do not seem reasonable to me and there are no references given to support the author's 'best guesses'.

- what do the chosen values for sigma in the simulation mean for the correlation between tests? How is it incorporated that not all tests are correlated with each of the other? How are the values specific for schistosomiasis given the reference mentioned?

Reviewer #4: Please see general comments

**Conclusions**

-Are the conclusions supported by the data presented?

-Are the limitations of analysis clearly described?

-Do the authors discuss how these data can be helpful to advance our understanding of the topic under study?

-Is public health relevance addressed?

Reviewer #2: (No Response)

Reviewer #3: (No Response)

Reviewer #4: Please see general comments

**Editorial and Data Presentation Modifications?**

Reviewer #2: (No Response)

Reviewer #3: (No Response)

Reviewer #4: (No Response)

**Summary and General Comments**

Reviewer #2: I have only three very minor edits, otherwise I am happy with the revision of this paper. The authors have taken the time to address all the comments and have done so thoroughly. It reads much more smoothly and I really appreciate the more apparent link to field study design. 

My three comments are: 

1. In the author summary, perhaps using numbers instead of semi-colons to separate out the list of study design issues. 

2. Line 49 I think should say "partial gold standard" instead of partially

3. Line 52 I think should say "two previous studies" not two previously studies.

Reviewer #3: - I do not believe that the corrections sufficiently adresses the major problems previously identified by multiple reviewers.

- There does not seem to be a gold standard diagnostic test for schistosomiasis and this is still one of the majore assumptions behind the manuscript.

- The used statistical models do not capture the real correlation structure between diagnostic tests for schistosomiasis and are not validated because the sample size is too small.

- Therefore, I do not have confidence that the simulation and conclusions from it are supported for schistosomiasis.

-250 people might be a typical size of a study to determine prevalence but it is simply too low to determine which is the appropriate model to use and to determine diagnostic accuracy of multiple diagnostic tests including their correlation structure.

Reviewer #4: In this manuscript, the authors describe and implement three latent class models, using two datasets (one of S. haematobium and one of S. Mansoni). The authors then discuss the limitations of such models to estimate disease prevalence and the issues with the different diagnostics.

I thoroughly enjoyed reading this manuscript, it is mostly clear and well written. The methods are appropriate and highlight some of the challenges of dealing with multiple diagnostics in the absence of a gold standard.

Overall, I feel this manuscript is almost ready for publication, I would like some points clarified further before this is acceptable for publication:

1. I feel the title can be misinterpreted. When I read it, I thought this manuscript was about issues with schisto studies, but really it is issues with Latent class models in general, using a schisto example to discuss it. Most recent schisto diagnostic studies (which are cited in the manuscript), do not use any of the three models described here. I would recommend the authors to consider an alternative title.

2. Following from the above, recent papers on schisto diagnostics use mechanistic or semi-mechanistic approaches, I would like to hear the thoughts of the authors in their discussion as the issues they highlight here could potentially be also in those models (or not)?

3. I would like the use of binary data discussed a little bit too, many diagnostics (specially in schisto) are not black/white. There is a lot of information that is potentially lost when making them binary (i.e. KK can give intensity, CCA is also somewhat semi-quantitative).

4. As the main purpose of the paper is to explain these Latent class methods, I worry a bit that the methods section itself is a bit too technical for a majority of the plos NTD audience. I’m not suggesting for the equations to be removed, but I leave it at the authors discretion if they feel some of the methodology text can be clarified further.

5. Following on the above, I want to commend the authors for the shiny App. I think it is a great addition to the paper and I take my hat off. It is a simple, yet effective way of getting the point across and I know they can be a bit fiddly to code.

6. Regarding the gold standard. It would be good if the assumption of a gold standard is discussed a bit more. What are the biases of assuming a diagnostic as a gold standard when it is not, as no diagnostic is perfect?

7. I was a bit surprised by the result that egg sensitivity was higher in the lower prevalence setting (Tables 1 & 3), as we would expect the opposite, a lower sensitivity with low prevalence. It has to be due to a difference between haematobium and mansoni, so potentially worth a sentence or two?

Additional thoughts:

Line 15. Where do the 6 false positives come from?

Line 51. What if DNA is not available? Does DNA necessarily imply live worms (for example after treatment)?

PLOS authors have the option to publish the peer review history of their article (what does this mean?). If published, this will include your full peer review and any attached files.

Reviewer #2: No

Reviewer #3: No

Reviewer #4: No
---

## [Editor Report · Decision Letter 2]

18 Dec 2020

Dear Dr Koukounari,

We are pleased to inform you that your manuscript 'Latent Class Analysis: Insights about design and analysis of schistosomiasis diagnostic studies' has been provisionally accepted for publication in PLOS Neglected Tropical Diseases.

Best regards,

Maria Elena Bottazzi

Deputy Editor

Maria Elena Bottazzi

Deputy Editor

---

## [Editor Report · Acceptance letter]

27 Jan 2021

Dear Dr Koukounari,

We are delighted to inform you that your manuscript, "Latent Class Analysis: Insights about design and analysis of schistosomiasis diagnostic studies," has been formally accepted for publication in PLOS Neglected Tropical Diseases.

Best regards,

Shaden Kamhawi

co-Editor-in-Chief

Paul Brindley

co-Editor-in-Chief
